# Dynamic composition of stress granules in *Trypanosoma brucei*

**Htay Mon Aye, Feng-Jun Li, Cynthia Y. He** *

Department of Biological Sciences, National University of Singapore, Singapore, Singapore

* dbshyc@nus.edu.sg

## Abstract

Stress granules (SGs) are stress-induced RNA condensates consisting of stalled initiation complexes resulting from translational inhibition. The biochemical composition and function of SGs are highly diverse, and this diversity has been attributed to different stress conditions, signalling pathways involved and specific cell types. Interestingly, mRNA decay components, which are found in ubiquitous cytoplasmic foci known as processing bodies (PB), have also been identified in SG proteomes. A major challenge in current SG studies is to understand the cause of SG diversity, as well as the function of SG under different stress conditions. *Trypanosoma brucei* is a single-cellular parasite that causes Human African Trypanosomiasis (sleeping sickness). In this study, we showed that by varying the supply of extracellular carbon sources during starvation, cellular ATP levels changed rapidly, resulting in SGs of different compositions and dynamics. We identified a subset of SG components, which dissociated from the SGs in response to cellular ATP depletion. Using expansion microscopy, we observed sub-granular compartmentalization of PB- and SG-components within the stress granules. Our results highlight the importance of cellular ATP in SG composition and dynamics, providing functional insight to SGs formed under different stress conditions.

## Author summary

Stress granules (SGs) and processing bodies (PBs) are cytoplasmic RNA granules containing multiple proteins with functions in translation initiation and mRNA degradation. mRNA and proteins can move between SGs, PBs and translating polysomes, likely regulating gene expression under different cellular conditions. However, little is known about the cellular factors regulating the dynamic behaviour of these RNA granules. In the unicellular Trypanosome parasites, SGs are formed under starvation stress and during differentiation. In this study, we found that in starved *T. brucei*, the composition and the dynamics of the SGs are distinct, depending on the supply of extracellular carbon sources and subsequent cellular energy (ATP) levels. Using expansion microscopy and super-resolution microscopy, functionally-distinct protein components are found in different sub-granular compartments within the SGs. Our findings emphasized the dynamic changes in

**Data Availability Statement:** The authors confirm that the data supporting the findings of this study are available within the article and its supplementary materials.

**Funding:** This work was supported by Singapore Ministry of Education (MOE-T2EP30221-0002 to

CYH and A-0008403-00-00 to CYH). The funder had no role in study design, data collection and analysis, decision to publish, or preparation of the manuscript.

**Competing interests:** The authors have declared that no competing interests exist.

SG composition, organization, and potential functions, which are affected by cellular ATP levels that can fluctuate under various stress conditions.

## Introduction

Stress granules (SGs) are membrane-less, liquid-like condensates induced by various stresses such as starvation, heat shock, oxidative stress, UV, and chemical treatments [1–3]. Under stress, apart from specific stress-response genes that are selectively translated [4], the bulk of mRNAs and proteins are released from translating polysomes, recruited by nucleating proteins that contain intrinsically disordered regions or low complexity domains, and packaged into SGs in a liquid-liquid phase separation (LLPS) process [5,6]. Cellular ionic strength and osmolytes that affect molecular interactions can also modulate LLPS and the formation of SGs [7–9].

Proteomics studies reveal a diversity in protein compositions among SGs induced by different signalling pathways in different cell types upon different stress conditions [10–13]. Interestingly, while SGs contain components of stalled translation initiation complexes, many mRNA decay components that are usually associated with processing bodies (PBs) have also been observed in SG proteomes [14–16]. The PBs are another type of RNA condensates that are constitutively present in cells [17,18], but can become more prominent during stress [19]. During glucose starvation in yeast, several mRNA decay components were found to be present in newly formed SGs but then dissociate from the mature SGs, supporting that PBs act as scaffold for the formation of SGs [14]. Arsenite-induced SGs in mammalian cells also appear to arise from PBs [20]. However, in mammalian cells under oxidative, osmotic stress conditions, SGs and PBs are found in close proximity with each other [15,21,22]. Exchange of protein and mRNA components between these structures is observed [15,23]. It is speculated that mRNAs meant for degradation are sorted in SGs before they are handed over to PBs for degradation [15,24]. Cellular factors that contribute to the dynamic and diverse protein makeup of SGs remain largely unknown.

*Trypanosoma brucei* is an early divergent eukaryote with unique features of mRNA metabolism and gene expression regulation. Unlike most other eukaryotes, transcription in *T. brucei* is polycistronic in nature and lacks introns and gene-specific promoters [25,26]. As such, regulation of gene expression occurs primarily at the post-transcriptional level, with mRNA decay and translation efficiency playing major roles [25,27,28]. Biomolecular condensates containing RNA and proteins with important roles in RNA metabolism have been observed during differentiation [29] and under stress conditions [30,31]. Stress-induced RNA granules in *T. brucei* display features characteristic of SGs. They contain components of translation initiation complexes such as poly-A binding protein (PABP), eIF4E and eIF4G. Additionally, their formation is dependent on global polysome collapse [30,32], suggesting that *T. brucei* SGs contain stalled initiation complexes just like other organisms.

On the other hand, *T. brucei* SGs also contain mRNA decay components such as DHH1 and SCD6, which are also components of PBs [32–34]. The DEAD box RNA helicase DHH1, a component of the mRNA decay pathway [35–37] is found present in the newly formed SGs upon starvation in phosphate buffered saline (PBS). But after prolonged starvation, DHH1 dissociates from the SGs [32] in a manner similar to that observed in yeast [14]. Considering that DHH1 is an ATPase, and its ATPase and ATP-binding activities have been shown to be crucial for its granule localization [38], we hypothesized that cellular ATP may have a role in regulating DHH1 association with the SG in *T. brucei*. As a parasitic organism, *T. brucei* relies on

extracellular carbon sources such as glucose and proline for cellular ATP production [39–41]. Previous work has shown that intracellular ATP can be rapidly modulated by altering the availability of carbon sources in the medium [42], making it feasible to study SG biology under different cellular ATP conditions. We found that SGs formed under ATP-competent and ATP-deprived conditions had different protein compositions and exhibited distinct dynamic properties. While some PB and SG components are stably associated with the SGs, several components dissociated from the SGs in response to depleted cellular ATP, correlating with changes in SG dynamics and cellular translation activities. Our results emphasize the importance of cellular ATP on SG composition and potential functions.

## Materials and methods

### Cell culture

Procyclic form (PCF) *T. brucei* with stably integrated pSmOxNus vector that supports both tetracycline- and cumate-inducible expression [42] were used for all experiments. The cells were maintained in Cunningham's medium supplemented with 10% heat-inactivated fetal bovine serum (Hyclone) and suitable selection drugs.

### Starvation conditions

To induce starvation, *T. brucei* PCF cells harvested at mid-log phase were washed once with PBS or gPBS (PBS containing 1 g/L or 5.6 mM glucose unless otherwise stated), resuspended and incubated in the same starvation medium at a concentration of $5 \times 10^6$ cells/ml at 27°C for specified periods. To deplete cellular ATP, 2DG (Sigma) was added to the starvation buffer (PBS2DG) at 5.6 mM (i.e. 0.9 g/L) final concentration. To deplete *T. brucei* of carbon sources while maintaining amino acid supply, an "energy-depleted medium" (ED) was prepared according to the formula shown in S1 Table. Cells were washed once, resuspended and incubated in ED medium for 2 h.

To study the effect of individual amino acid on SG formation during starvation, each individual amino acid was added to PBS at concentrations present in the Cunningham medium (see S1 Table) with pH adjusted to 7.2. To increase the osmolarity or ionic strength, NaCl (50 mM or 100 mM final concentration) was added to the starvation buffer; and to decrease the osmolarity, the starvation buffer was diluted with distilled water at 1:1 or 2:1 (vol:vol) ratio. *T. brucei* were washed once, resuspended and incubated in the respective starvation buffers for specified periods.

### Plasmids

All plasmids used in this study are summarised in S2 Table. Plasmid construction was performed using traditional restriction enzyme-based cloning or Gibson assembly [43,44]. Genomic sequences were retrieved from the Tritryp Database (http://tritrypdb.org/tritrypdb/).

### ATP measurements

The boiling water method [45] was used to extract cellular ATP. Following respective starvation treatments, $2 \times 10^7$ cells were pelleted and resuspended in 100 μL cold $dH_2O$. The samples were then vortexed for 1 min and heated at 99 °C for 10 mins. The lysates were centrifuged at 17,000 g for 5 min at 4 °C, and the supernatants were collected for ATP measurements using an ATP Assay kit (BMR, Cat #A-107) following manufacturer's protocol. Relative Light Unit (RLU) was recorded using dual luciferase assay settings on a Hidex Sense microplate reader (Hidex Oy).

## Fluorescence microscopy

Cells were washed and resuspended in serum-free medium (as control) or respective starvation medium. Cells attached to coverslips were fixed with 4% paraformaldehyde in PBS for 20 mins at room temperature and permeablized with 0.25% Triton X-100 in PBS for 10 min. DAPI (2 µg/ml) was used to stain nuclear and mitochondrial DNA. For hypotonic lysis of stress granules, *T. brucei* starved in PBS or gPBS for 2 h were harvested, washed once with PBS, resuspended in ice-cold hypotonic cell lysis buffer [32] and incubated on ice for 5 min. Reporter-tagged proteins were visualized by direct fluorescence (mScarlet or mNeonGreen) or indirect immunofluorescence using anti-HA antibodies (mouse mAb sc-7392 from Santa Cruz or rat mAb 7C from ChromoTek) followed with appropriate secondary antibodies.

Live cell imaging was performed on cells immobilized by agarose gels. Procyclic cells were washed and resuspended in PBS or gPBS to induce starvation. Cells were spread on the surface of 1% agarose gel prepared in PBS or gPBS. The gel was sliced and mounted onto an imaging dish with the cell-side facing the glass coverslip bottom. Images were acquired at room temperature using an Axio Observer Z1 fluorescence microscope with a 63× (NA1.4) objective (Carl Zeiss) and a CoolSNAP HQ2 CCD camera (Photometrics). Images were acquired using ZEN imaging software (Zeiss) and processed using Fiji. For quantifications, number of cells containing PABP2-labelled puncta (SGs) in a population of at least 200 cells were counted at each time point. The results are shown as percentage of cells with SG.

For STED super-resolution microscopy, immunofluorescence labelling was performed using anti-HA (mouse mAb sc-7392 from Santa Cruz or rat mAb 7C from ChromoTek) for HA-tagged proteins, mouse mAb anti-TY [46] for TY-tagged proteins or anti-RFP (rat mAb 5F8, ChromoTek) for mScarlet. For secondary antibody labelling, Abberior Star Red and Star Orange antibodies were used. STED imaging was performed using a STEDYCON nanoscope (Abberior) equipped with 561nm and 649nm laser. A pulsed STED laser (775 nm) was used for depletion.

## Expansion microscopy

The protocol was adapted from [47]. Following immunostaining with anti-HA, anti-RFP or anti-TY, or RNA in situ hybridization with oligo d(T) (see below), cells were incubated in 0.25% glutaraldehyde in PBS for 10 mins at room temperature followed by washing three times with PBS. Coverslips were then equilibrated by incubation in monomer solution (2 M NaCl, 2.5% acrylamide, 0.15% N,N'-methylenebisacrylamide, 8.625% sodium acrylate, 1x PBS) for 1 min (cell side facing down on the monomer solution on the parafilm in a humid chamber on ice). Gel polymerization was initiated by incubation in monomer solution containing 0.2% ammonium persulfate (APS) and 0.2% tetramethylethylendiamine (TEMED) for 30 min at room temperature. After gelation, solidified gel was removed from coverslips and incubated in digestion buffer (0.5% Triton X, 0.8 M guanidine HCl, 8 units/mL Proteinase K, 1× TAE) at 37 ˚C for 30 min. Following protease digestion, gels were incubated in deionized water to allow expansion. DAPI (2 µg/ml) was added to the water during the first expansion step to stain nuclear and mitochondrial DNA. Water was exchanged every 30 min for a total of 6 times or every 30 min for 2 times followed by overnight incubation in water.

## RNA *in situ* hybridization

*T. brucei* starved in PBS or gPBS were fixed with 4% paraformaldehyde for 15 min at room temperature. Cells were washed in PBS, resuspended and attached to coverslips. Coverslips were washed with 25 mM $NH_4Cl$ for 10 min. This is followed by cell permeabilization and blocking in PBS solution containing 0.5% saponin and 2% BSA for 1 hr. Cells were then

incubated with pre-hybridization buffer (2% BSA, 5xDenhard's solution, 4xSSC, 5% Dextran sulfate, 35% deionized formamide, 0.5 µg/µl Torula yeast RNA, 10 U/ml RNase inhibitor) in a humidified chamber for 2 h and then in pre-hybridization buffer containing 2ng/µl biotiny-lated oligo d(T) probe (Promega) for 12 h at room temperature. Cells were washed twice in 4xSSC buffer (0.15M sodium chloride and 15mM sodium citrate), once in 2xSSC buffer and once in PBS. This is followed by blocking in 3% BSA and incubation in Alexa-Fluor-568–strep-tavidin (Invitrogen; 1:200).

### HPG labelling, click reaction and in-gel fluorescence detection

A total of $4\times10^8$ *T. brucei* were pre-starved in gPBS or PBS for 15 min to deplete the intracellular methionine. 50 µM Click-iT HPG (L-Homopropargylglycine, Invitrogen) was then added and the cells were incubated for another 2 hrs at 27 ˚C. After HPG labeling, cells were harvested and lysed with 0.16% SDS supplemented with 1× Halt Protease Inhibitor Cocktail (Pierce), 50 µg/ml RNase and 50 µg/ml DNase. The suspension was sonicated briefly to facilitate cell lysis, and the cell lysates were cleared by centrifugation. For click reactions, the supernatant containing HPG-labelled proteins were incubated with a cocktail of 10 µM Alexa Fluor 488 azide (Alexa Fluor 488 5-carboxamido-6(azidohexanyl), bis(triethylammonium salt), Invitrogen), 1 mM Tris (2-carboxyethyl) phosphine (TCEP) 100 µM Tris [(1-benzyl-1H-1,2,3-tria-zol-4-yl)methyl, Sigma] amine (TBTA ligand, Sigma) and 1 mM $CuSO_4$. The samples were incubated for 2 h at room temperature or overnight at 10 ˚C shielded from light with gentle mixing. Ice-cold acetone was then added to the samples in a ratio of 4:1 (v/v) and the mixture incubated for another 30 min at -80 ˚C to precipitate the clicked proteins. The proteins were centrifuged at 14,000 rpm for 20 min at 4 ˚C and air-dried. The protein precipitates were then dissolved in 1×SDS loading buffer and separated by 10% SDS-PAGE. The gel was imaged using a Typhoon 9410 laser scanner (GE) and analyzed with ImageQuant software.

### Cell death assay

Cells were cultivated in medium or treated in different starvation buffers for indicated times. Cell were then stained with 2 µM Propidium iodide (PI) for 15 min and washed once with the same medium or buffers. Afterwards, cells were fixed with 4% PFA and washed three times with PBS before flow cytometry analysis was performed. Cells without PI-staining were used as negative control and cells permeabilised by 1% NP-40 and stained with PI were used as positive control. The PI intensity was measured using Beckman Coulter CytoFLEX LX B-R-V-Y-U-I flow cytometer and data was analysed using free online application Floreada.io (https://floreada.io).

## Results

### Stress granules are formed in both cellular ATP-deprived (PBS) and cellular ATP-competent (gPBS) stress conditions in *Trypanosoma brucei*

Procyclic *T. brucei* cells cultivated in Cunningham medium have a stable cellular ATP concentration in the range of 4–5 nmol/$10^7$ cells [42]. Upon starvation in PBS, cellular ATP declined continuously, to ~50% of the starting level at 30 min and plateaued at ~10% after 60 min (Fig 1A). Inclusion of 1 g/L glucose in PBS (gPBS) stabilized cellular ATP until at least 2 h post starvation (Fig 1A). These patterns are similar to those previously described [42,48]. PBS with and without glucose thus provide a useful system to modulate cellular ATP and examine its effects on SG biology over the course of starvation stress.

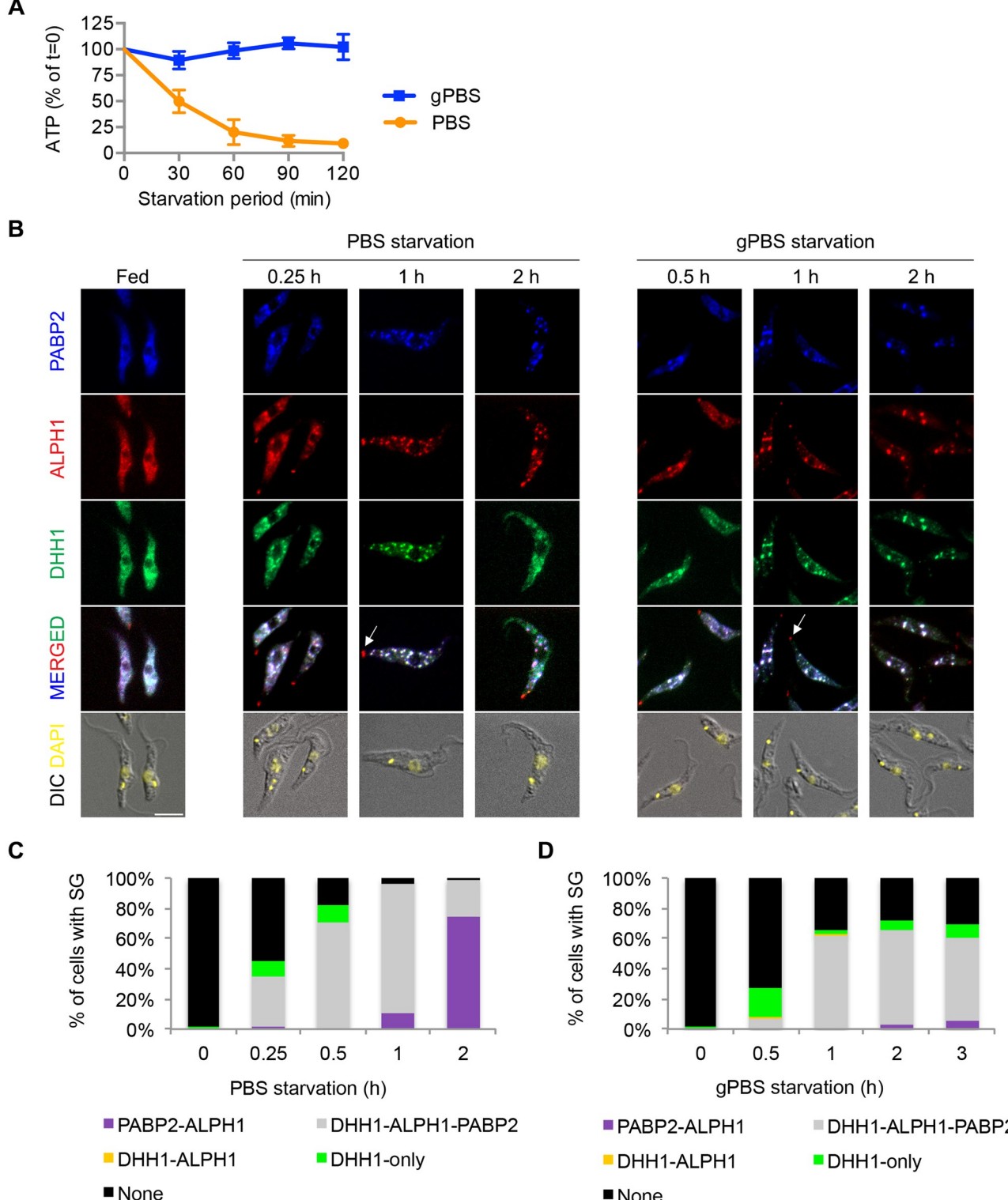

**Fig 1. SGs are formed in both PBS and gPBS.** Cells stably expressing PABP2-HA, ALPH1-mScarlet and mNeonGreen-TY-DHH1 were starved in PBS and gPBS for 2 and 3 h, respectively. Cells were analysed for cellular ATP (**A**) and SG markers (**B**) at indicated times. The cells were classified based on the presence of SG and SG markers and the quantitation results are shown in (**C**) and (**D**). None, cells that did not contain detectable SGs; DHH1-only, cells containing granules marked by DHH1 only; PABP2-ALPH1, DHH1-ALPH1, and PABP2-ALPH1-DHH1, cells containing SGs marked by indicated markers. ALPH1-mScarlet and mNeonGreen-TY-DHH1 were visualized by direct mScarlet and mNeonGreen fluorescence. PABP2-HA was

detected by immunofluorescence with anti-HA. Arrows mark the posterior puncta containing ALPH1 only. At least 200 cells were counted at each time point. Scale bar: 5 μm.

Poly-A binding protein 2 (PABP2) and DHH1 have been used to visualize stress-induced granules in *T. brucei* [32]. PABP2, a component of the translation initiation complex, is considered a canonical SG marker; DHH1 however, can be found in both PB and SG [49,50]. Recently an ApaH-like phosphatase (ALPH1) is identified as a major mRNA decapping enzyme in *T. brucei* and it is present in the SGs [51]. We generated cells stably expressing PABP2-HA, ALPH1-mScarlet and mNeonGreen-TY-DHH1 from their native alleles. All 3 markers showed cytoplasmic distribution in parasites cultivated in full medium (Fig 1B). Upon PBS starvation for 1 h, all 3 markers became enriched in SGs (Fig 1B and 1C). Additional ALPH1 puncta were found at the posterior pole of the starving cells (arrows, Fig 1B), consistent with previous observations [51]. These posterior puncta are unlikely SGs, as they do not contain DHH1 [51] or PABP2 (Fig 1B). Stable expression of PABP2-HA, ALPH1-mScarlet and mNeonGreen-TY-DHH1 markers under fed or starvation conditions at the expected molecular weights was confirmed by immunoblots (S1A and S1B Fig).

Deprivation of both amino acids and carbon sources is thought to be required for SG formation in *T. brucei* [30]. However, SGs with bright and discrete tri-colour labelling were also observed in cells starved in gPBS (Fig 1B and 1D). When cellular ATP was depleted by preincubation in medium without carbon source (ED) prior to PBS starvation, or by treatment with 2-deoxyglucose (2DG), a glucose analog and a glycolysis inhibitor [52] at the time of PBS starvation (PBS2DG), SG formation was completely inhibited (S2 Fig). These results suggest that the presence of carbon sources and cellular ATP are important for SG formation in *T. brucei*. During PBS starvation, the presence of low cellular ATP at ~50% of the starting level up to 30 min (Fig 1A) likely supports SG formation in the early phase of the treatment. No significant cell death was observed for any of the above starvation or ATP depletion conditions (S2D Fig).

Although SGs were formed in both PBS and gPBS, differences in formation kinetics and efficiency were noted between the two conditions (Fig 1B–1D). In PBS, SGs were observed as early as 15 min post starvation in ~35% of cells. Within 1 h starvation, >95% of cells contained SGs (Fig 1B and 1C). In gPBS, SG formation appeared delayed, with ~5% and 55% of cells containing SGs at 30 min and 1 h post-starvation, respectively. By 2 h, ~70% of cells contained detectable SGs. This percentage did not further increase upon prolonged starvation (Fig 1B and 1D).

DHH1 also exhibited different behaviour in PBS and gPBS-starved cells. During PBS starvation, granules containing DHH1 only were observed in the early starvation period, followed by a rapid increase of SGs containing all three markers (PABP2-ALPH1-DHH1). However, at ~1 h post-starvation, DHH1 became diffused and was no longer enriched in SGs in some cells. At 2 h post-starvation, ~84% of the cells contained SGs with PABP2 and ALPH1, but not DHH1 (Fig 1B and 1C). Live cell imaging confirmed the dissociation of DHH1 from SGs late in PBS starvation (S3 Fig). The behaviour of DHH1 in PBS-starved cells—early appearance in SGs and later dissociation from SGs–generally agrees with previous report [32]. In gPBS however, DHH1 remained associated with the SGs as bright and discrete puncta throughout the course of starvation (Fig 1B and 1D).

## A subset of proteins dissociate from SGs upon cellular ATP depletion

Given the characteristics of DHH1 as an ATPase-containing DEADbox helicase [38], we reasoned that DHH1 dissociation from SGs may be caused by reduced cellular ATP levels during

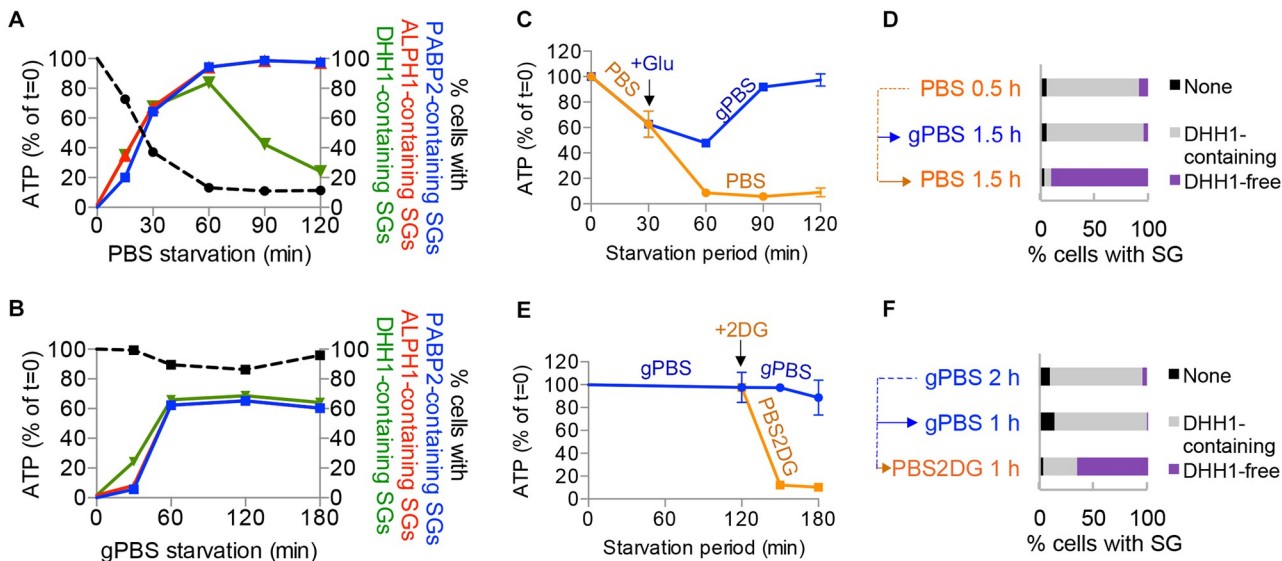

**Fig 2. Cellular ATP modulates DHH1 dissociation from SGs. A-B**) Proportion of cells with PABP2-, ALPH1- or DHH1-containing SGs were plotted against cellular ATP, using the same data from experiments shown in Fig 1. **C-D**) Cells were starved in PBS for 30 mins followed by incubation in PBS or gPBS for an additional 90 mins. **E-F**) Cells expressing PABP2-mScarlet and mNeonGreen-DHH1 were starved in gPBS for 2 h followed by incubation in gPBS or PBS2DG for an additional 1 h. Cellular ATP was measured at specified time points for each starvation condition. Cellular ATP was measured at specified time points for each starvation condition (**C**) and (**E**). Percentage of cells containing specified SG profile was quantified at the end of the starvation schemes (**D**) and (**F**). None, cells that did not contain detectable SGs; DHH1-containing, cells containing SGs marked by DHH1; DHH1-free, cells containing SGs marked by PABP2 and ALPH1 only. ALPH1-mScarlet and mNeonGreen-TY-DHH1 were visualized by direct mScarlet and mNeonGreen fluorescence, respectively. At least 200 cells were counted at each time point.

starvation. The experiments shown in Fig 1 were replotted, showing the percentage of cells with DHH1, PABP2 or ALPH1-containing SGs versus cellular ATP during the course of starvation (Fig 2A and 2B). In PBS, the proportion of cells with DHH1-containing SGs began to decline after 1 h starvation, when cellular ATP was ~10% of the starting level. However, the proportion of cells with PABP2- or ALPH1-containing SGs remained unchanged from 1 h onwards. In gPBS, both cellular ATP and the proportion of cells with DHH1-containing SGs remained constant after 1 h. DHH1 dissociation with SGs is thus temporally correlated with reduced cellular ATP during PBS starvation.

To further examine if reduced cellular ATP could induce DHH1 dissociation from SGs, cells were starved in gPBS for 2 h and then starved in PBS2DG for an additional 1 h. The presence of 2DG led to rapid reduction of cellular ATP to about ~10% of the starting level (Fig 2E), and increase of cells containing DHH1-free SGs (Fig 2F). As new SG cannot form in PBS2DG (see S2 Fig), the DHH1-free SGs observed in this experiment could only result from DHH1 dissociation from existing SGs. As control, glucose (1 g/L final concentration) was added to PBS at 30 min post-starvation when the cellular ATP was ~50% of the starting level. With the addition of glucose, cellular ATP did not drop further and DHH1 was retained in the SGs (Fig 2C and 2D). The addition of glucose thus stabilized cellular ATP as well as DHH1 association with the SGs. Altogether, these findings demonstrated that DHH1 dissociates from SGs in response to reduced cellular ATP; PABP2 and ALPH1 however, are stably associated with SGs regardless of cellular ATP conditions.

In addition to PABP2, ALPH1 and DHH1, several other SG components have been identified in previous proteomics analyses [32]. Selected candidates with confirmed SG localizations [32,53] were endogenously tagged with mScarlet in cells stably expressing mNeonGreen-

TY-DHH1 or PABP2-TY-GFP (Fig 3). The SG association of each selected component was then monitored by fluorescence microscopy in cells starved in gPBS for 2 h, and PBS for 0.5 h and 2 h, where the cellular ATP was at 100%, ~50%, and ~10% of steady levels. As shown in Fig 3, four SG components SCD6, RBP6, eIF4E1 and CAF1 showed DHH1-like behaviour: they were discretely present in the SGs at 0.5 h PBS-starvation as well as 2 h gPBS-starvation, but were diffused at 2 h PBS-starvation. On the other hand, eIF4G1 and ALPH1 displayed PABP2-like behaviour and were localized to the SGs under all three starvation conditions. These results indicated that cellular ATP affects protein composition of the SGs. As many of these proteins have known functions in translation initiation or mRNA decay, their presence/absence in SG may affect mRNA fate and SG function under different stress conditions.

## Cellular ATP affects dynamic exchange between SGs and translating polysomes

SGs are highly dynamic structures, constantly recruiting, sorting, and directing mRNA-protein complexes (mRNPs) to-and-from different cellular components. Upon translational arrest, mRNAs are released from polysomes and recruited to SGs [15,20,54]. To examine SG dynamics during translational arrest in *T. brucei*, translation inhibitors cycloheximide and puromycin were added to cells expressing PABP2-mScarlet and mNeonGreen-TY-DHH1, either at the same time of starvation before SG formation, or 2 h post-starvation when SGs had formed (Fig 4A). Cycloheximide inhibits translation elongation and "preserves" translating mRNAs in the polysomes [20], whereas puromycin causes premature termination of translation and releases mRNA and ribosomes from the translating polysomes [7].

During co-starvation treatment, cycloheximide inhibited SG formation in both PBS and gPBS (Fig 4B and 4C), supporting that the mRNP contents of SGs are derived from polysomes. Interestingly, puromycin inhibited SG formation in gPBS, but not in PBS (Fig 4B and 4C), suggesting a requirement of active translation in gPBS-induced SG formation. During post-starvation treatment, the percentage of cells containing SGs (labelled by PABP2-mScarlet) decreased upon the addition of either cycloheximide or puromycin in gPBS. As both drugs inhibited new SG formation in gPBS, the reduction of existing SGs is likely due to dynamic out-flow of SG materials. In PBS however, existing SGs persisted through both cycloheximide and puromycin treatments (Fig 4D and 4E), suggesting a lack of dynamicity. Importantly, while protein synthesis appeared completely inhibited in PBS, a low level of protein synthesis was still observed during gPBS starvation (Fig 4F). The dynamic in-flow and out-flow of materials in gPBS-induced SG is thus consistent with the presence of active translation in gPBS-starved cells.

In a previous study [32], PBS-induced SGs are isolated by incubating cells with a hypotonic buffer, which permeates the cell membrane but maintains the integrity of the subpellicular microtubules, thereby enriching the SGs within the microtubule "cage". The same lysis method was however, unable to preserve SGs formed in gPBS (S4 Fig), further supporting that SGs formed in PBS are more static and stable than SGs formed in gPBS.

## Components of PB and SG are organized into different compartments within the granule

Protein components of SG and PB are found co-localized in starvation-induced SGs in *T. brucei* (28; this study). This observation is similar to that reported in glucose starvation-induced stress granules in yeast [14], but distinct to mammalian cells where SG and PB components are usually found adjacent to each other [15,21,22]. As SGs and PBs are considered to be compositionally and functionally distinct organelles [18,49,55], it was surprising that their components co-localise to the same granules in yeast and in *T. brucei*. Expansion microscopy (ExM)

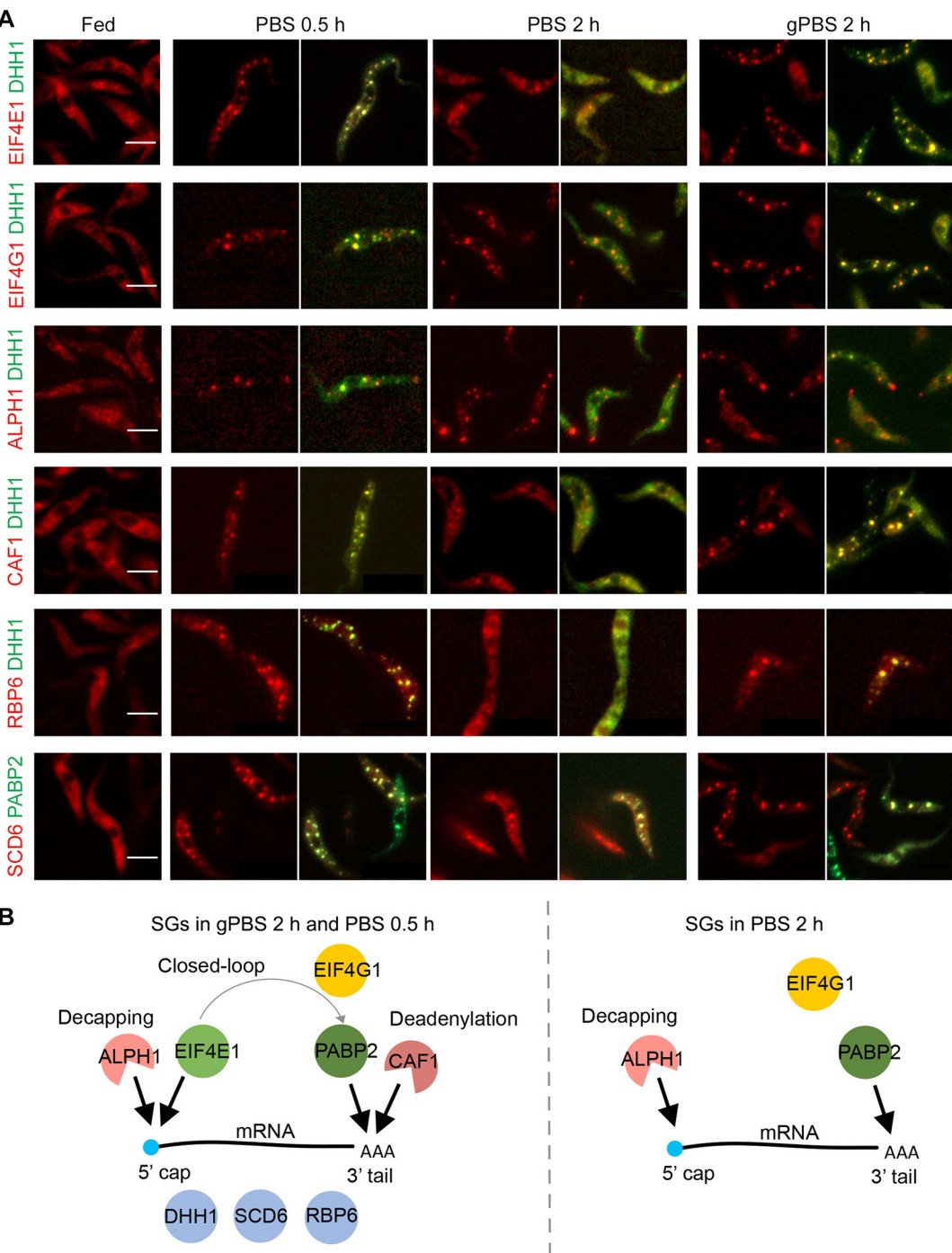

**Fig 3. A subset of SG proteins dissociates from SGs in response to cellular ATP depletion. A)** Selected SG proteins were endogenously tagged with mScarlet and expressed in cells stably expressing mNeonGreen-TY-DHH1 or PABP2-TY-GFP. SG localization of the mScarlet-labelled proteins was monitored under specified stress conditions. **B)** Schematics showing components found in gPBS- or PBS-induced SGs 2 h post starvation. Scale bar: 5 μm.

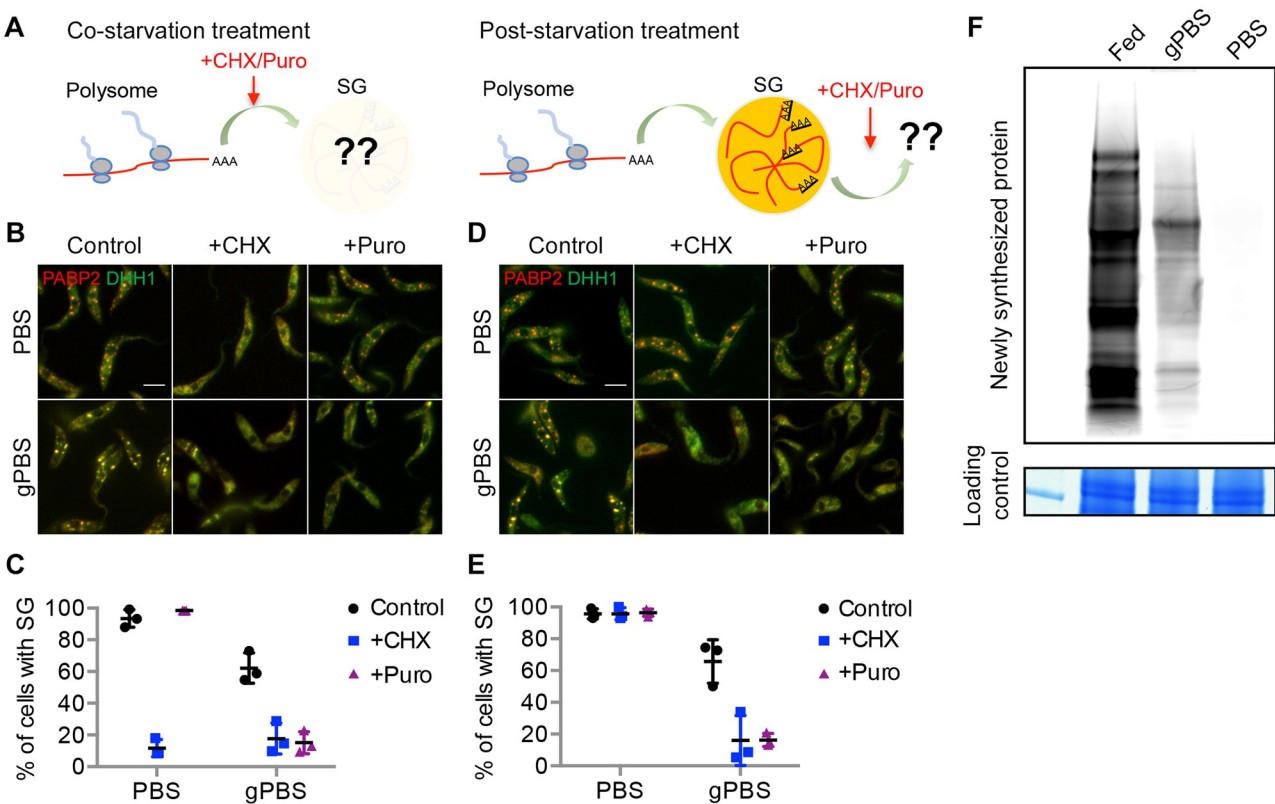

**Fig 4. gPBS, but not PBS, supports dynamic exchange of mRNPs between SGs and polysomes. A)** Experimental setups for co-starvation treatment to monitor mRNP recruitment to SGs during SG formation, and post-starvation treatment to monitor dynamic out-flow of SG materials. **B-C)** During co-starvation treatments, cells expressing PABP2-mScarlet and mNeonGreen-DHH1 were starved in PBS or gPBS for 2 h in the presence or absence of cycloheximide (50 μg/mL) or puromycin (200 μg/mL). **D-E)** During post-starvation treatments, cells expressing PABP2-mScarlet and mNeonGreen-DHH1 were starved in PBS or gPBS for 2 h, followed by the addition of cycloheximide or puromycin. Percentage of cells containing SGs was quantified in (**C**) and (**E**). Results shown are from 3 independent experiments. Scale bar: 5 μm **F)** Newly synthesized proteins were labelled with L-Homopropargylglycine (HPG), in cells cultivated in medium (fed), and starved in gPBS or PBS. HPG-labelled nascent proteins were fractionated by SDS-PAGE and visualized by click reaction with Alexa Fluor 488 azide. Equal loading was shown by Coomassie blue staining.

was therefore used to better understand the organization of different protein components in PBS- and gPBS-induced SGs. A protease-based ExM method [47] was used for our assays, as the conditions are compatible with all antibodies and probes used for the labelling of the SG components and poly-A RNA.

Under both PBS and gPBS conditions, the canonical SG marker PABP2 was found to be present in the SGs (Fig 5). Interestingly, upon an analysis of more than 100 PABP2-labelled granules, ~80% showed hollow regions at the centre or off to one side. Other known SG components eIF4G (Fig 5D) and polyadenylated RNA (Fig 5H) showed similar distribution and overlapped with PABP2 signals at ExM resolution. Interestingly, the decapping enzyme ALPH1 was found present in the hollow region, with little overlap with the PABP2 signal (Fig 5A and 5B). This organization was not affected by swapping the reporter tags on ALPH1 and PABP2 (Fig 5C), ruling out potential artefacts caused by different antibodies. This sub-granular compartmentalization of PABP2 and ALPH1 was also confirmed by STED super-resolution microscopy (S5 Fig), eliminating possible sample preparation artefacts during ExM. It was noted that the samples processed for FISH with oligo-d(T) prior to expansion produced more diffused and weaker signals for PABP2 (Fig 5H). This may be due to the harsher treatment conditions used for FISH that may affect the signal preservation of SG protein markers.

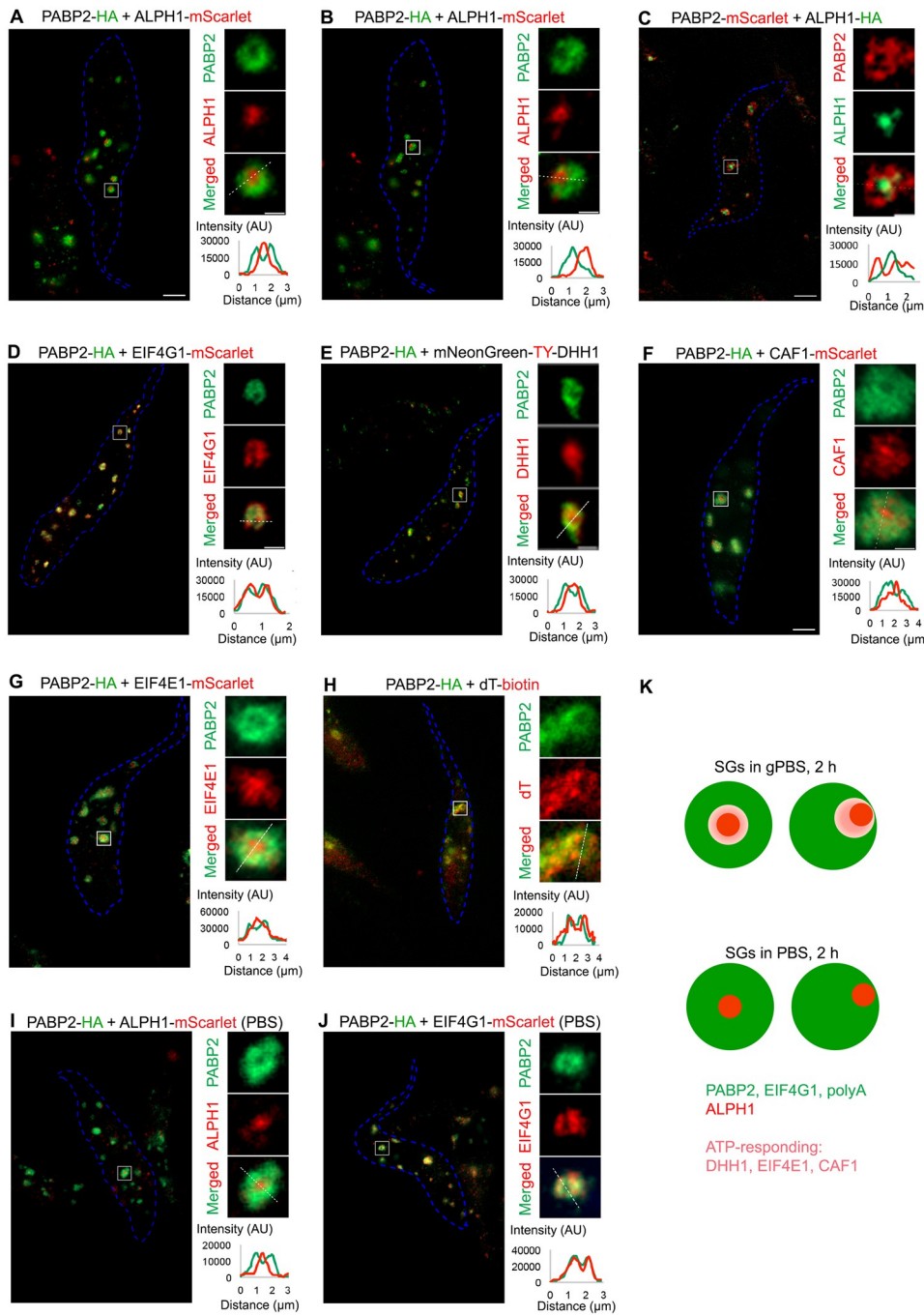

**Fig 5. ExM reveals SG and PB components in different compartments.** Cells stably expressing PABP2 and other SG components fused to indicated reporter tags were starved with gPBS (**A-H**) or PBS (**I, J**) for 2 h and processed for ExM. The SG components were visualized using antibodies specific to the reporter tags using anti-HA, anti-TY, anti-RFP (for mScarlet), or oligo d(T) for poly A-mRNA. Cell outline is marked in blue. Individual granules are selected for zoomed display to the right of each panel. Scale bar in the zoomed images: 1 μm. Fluorescence intensity profiles of PABP2 and proteins of interest across the granules (white dotted lines) are shown at bottom right of each panel. **K**) Schematic presentation of the sub-organellar compartmentalization of SG and PB components in stress granules formed in gPBS and PBS (2 h), respectively.

DHH1, a protein found in both PB and SG, showed similar localization pattern as ALPH1 in gPBS-starved cells (Fig 5E). However, compared to ALPH1, DHH1 signal partially over-lapped with PABP2. Other proteins common to both PB and SG, such as eIF4E1 and CAF1, showed similar localisation pattern to DHH1 (Fig 5F and 5G) with varying degrees of overlap with PABP2. In summary, these observations suggested compartmentalization between ALPH1 and other SG-specific components PABP2, eIF4G and polyadenylated RNA, all of which were stably associated with the SGs in both PBS and gPBS starvation conditions. Components that are found in both SGs and PBs (e.g. DHH1, eIF4E and CAF1) and disassociate from SGs in response to reduced cellular ATP, appeared to reside in a region overlapping both ALPH1 and PABP2 (Fig 5K).

## Osmotic property of external carbon sources can interfere with SG formation

Our results highlighted the importance of carbon source in cellular ATP, SG formation, SG protein composition and dynamic properties during amino acid starvation. In addition to glucose, procyclic *T. brucei* can also use proline as a carbon source for cellular ATP production [40]. Previous work [42] has shown that amino acid starvation in the presence of proline maintains cellular ATP. Surprisingly, SG was not observed in cells starved in proPBS, which contained 1 g/L proline (Fig 6A). Proline and alanine have been shown to be involved in osmoregulation in trypanosomatids [56], raising an interesting possibility that proline acts as an osmolyte, reducing SG formation through inhibition of LLPS [7].

We hence screened all amino acids that are present in *T. brucei* culture medium for their effects on SG formation in PBS. Of the 21 amino acids tested, the presence of proline (1 g/L), alanine (1.59 g/L), glycine (0.12 g/L), cysteine (0.08 g/L) or serine (0.2 g/L) impaired SG formation in PBS (S6 Fig). All five amino acids are relatively small and neutral, with the potential to act as organic osmolytes [57–59]. As controls, upon supplementation of PBS with 50 mM or 100 mM NaCl, which induces hyperosmolarity [60], drastically more SGs were observed in each *T. brucei* cell (S7 Fig). On the other hand, incubation in 0.33x or 0.5x PBS reduced SG formation (S7 Fig). These results supported an LLPS process leading to SG assembly in *T. brucei*.

In addition to amino acids, sugars such as glucose are also osmolytes that can potentially hinder SG formation. In a previous study, SG formation is not observed in PBS supplemented with 2.3 g/L glucose [30]. We compared the cellular ATP level and SG formation in PBS containing 1 g/L and 2.3 g/L of glucose. Despite similarly high cellular ATP levels, SG formation was much reduced in 2.3 g/L glucose (Fig 6B). It is possible that beyond the concentration at which cellular ATP saturates, the osmolyte property of glucose hinders SG formation. These

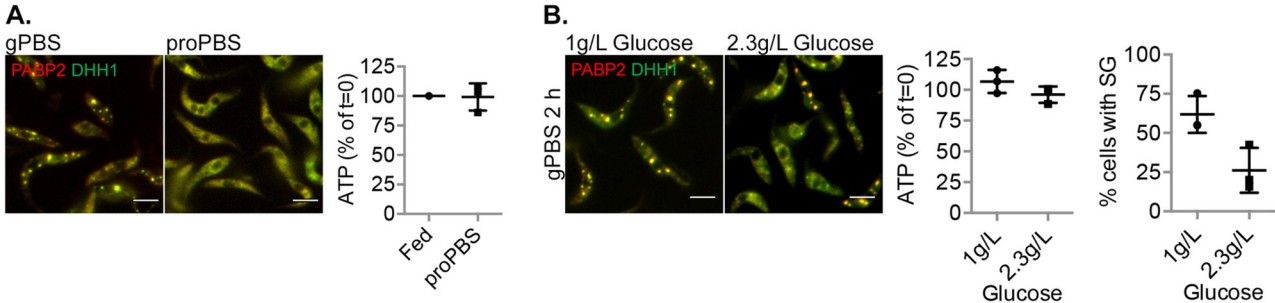

**Fig 6. Proline and glucose reduces SG formation in PBS.** Cells expressing PABP2-mScarlet and mNeonGreen-DHH1 were incubated for 2 h in PBS containing 1 g/L proline (proPBS) (**A**) or glucose at 1 g/L or 2.3 g/L (**B**). Cells were analysed for cellular ATP levels and presence of SG. Scale bar: 5 μm. At least 200 cells were counted for each starvation condition.

results also suggested that SG formation is a compound effect of multiple cellular factors, further attesting to the complexity of these organelles.

## Discussion

SG composition is known to be highly diverse depending on the stress conditions and cell types, and highly dynamic in material exchange with other cellular compartments. In this study, using *T. brucei* as a model, we compared SGs formed in cellular ATP-competent and -deprived conditions. Based on their behaviour upon ATP depletion, the SG proteins could be divided into two groups (c.f. Fig 3B). Group I contained ALPH1, PABP2 and eIF4G1, which were stably associated with the SGs regardless of cellular ATP levels. Group II proteins were associated with the SGs when cellular ATP was abundant (at least 50% of steady level), but dissociated from the SGs as cellular ATP dropped further. Other than RBP6, which is a *T. brucei*-specific protein, all Group II proteins are also PB components that have been observed to dissociate from mature SGs in yeast during glucose starvation [14]. In yeast, this dissociation behaviour has been interpreted as maturation of SGs [14].

While SG biogenesis and maturation in *T. brucei* may be similar to that observed in yeast [14], our results demonstrated that Group II proteins dissociated from the SGs in response to low cellular ATP, and this dissociation was correlated with changes in SG dynamics and protein translation activity in the starved cells. Group II protein DHH1 has been shown to be important for SG dynamics [61,62]. While the ATP binding of DHH1 is required for its assembly into PBs, the ATPase activity is required for disassembly/release of the granules [38]. As such, the cycle of ATP binding and hydrolysis enables the dynamic behaviour of DHH1. Upon ATP depletion during starvation, this dynamic recycling of DHH1 is likely compromised, resulting in their cytoplasmic localisation. It is also possible that DHH1 and other Group II proteins have roles in facilitating the dynamic exchange of materials between the SGs and the translating polysomes as observed in this study (c.f. Fig 4), and maintaining a low level of protein expression during the stress. It shall be noted that with the exception of DHH1, no other Group II proteins have known ATPase domain or ATP binding activities. How their activities and SG localization might be regulated by ATP is not clear.

It is worth noting that several Group II proteins including DHH1, SCD6 and CAF1 have known functions in protein translation repression and mRNA decay [34–36,63]. Whether or not they can inhibit translation and facilitate mRNA degradation within the SGs is unclear and yet to be tested experimentally. Given their ATP-dependent SG association, it is tempting to speculate that under ATP-competent stress conditions, SGs may function as a reservoir to sequester these mRNA decay proteins from the cytosol, enabling low level of protein translation on the polysomes.

Further functional and structural studies require understanding of the protein and RNA components of SGs formed under different conditions. As shown in this study, purification of SGs using conventional cell fractionation methods is likely challenging due to the highly dynamic nature of SGs formed under normal cellular ATP conditions. Proximity-based interactome methods may be more suitable. Given possible lack of cellular ATP in some of the stress conditions, the ascorbate peroxidase (APEX)-based proximity labelling methods [64,65] that can label both protein and RNA components in an ATP-independent manner will be useful.

Cellular ATP is a universal energy currency with important roles in various cellular processes [66–71]. In recent years, the novel role of ATP as a biological hydrotrope [72] has also boosted interest in its functions in LLPS, a common principle underlying the organization of membrane-less organelles such as the SGs [73]. While SG formation is generally thought to be

triggered by low cellular ATP [74,75], SG assembly and dynamics are both ATP-driven processes [62,76]. Our study highlights the importance of cellular ATP in the formation and dynamicity of the SGs, and the maintenance of a low level of protein expression in starved *T. brucei*. Our previous work [42] shows that cellular ATP is also required for autophagy, which functions in maintaining cellular ATP and amino acid supply under starvation stress [77–79]. As a parasitic organism, *T. brucei* lives in host environments rich in carbon sources, such as glucose in the blood and proline in the insect host [39–41]. The continuous presence of carbon source and steady cellular ATP supply may enable a complex, dynamic landscape of response pathways against different stresses such as amino acid starvation, host immune challenges, differentiation and temperature fluctuations. How different stress response pathways work together in shaping cellular responses and determining cell fate will be of particular interest in future work.

## Supporting information

**S1 Table. Composition of medium used in this study.**
(PDF)

**S2 Table. List of plasmids used in this study.**
(PDF)

**S1 Fig. Immunoblots of endogenously-tagged PABP2, ALPH1 and DHH1 under fed and starvation conditions.**
(TIF)

**S2 Fig. Depletion of cellular ATP alone does not induce SG formation.**
(TIF)

**S3 Fig. Temporal analysis showing DHH1 dissociation from SG during prolonged PBS starvation.**
(TIF)

**S4 Fig. SGs in PBS-starved cells are more stable.**
(TIF)

**S5 Fig. STED imaging confirms SG and PB components in different sub-organellar compartments.**
(TIF)

**S6 Fig. Several small and neutral amino acids impair SG formation in PBS.**
(TIF)

**S7 Fig. Altering ionic concentration of PBS affects SG formation.**
(TIF)

## Acknowledgments

The authors wish to thank Philippe Bastin for the anti-TY antibodies, and Dr. Zheng Chao and Mr. Goh Wei Jia from Abberior for assistance with STED super-resolution microscopy. We also thank Dr. Karthika Balasubramaniam for construction of pMO-mNeonGreen-Ty-DHH1 plasmid.

## Author Contributions

**Conceptualization:** Htay Mon Aye, Cynthia Y. He.

**Formal analysis:** Htay Mon Aye.

**Funding acquisition:** Cynthia Y. He.

**Investigation:** Htay Mon Aye, Feng-Jun Li.

**Methodology:** Htay Mon Aye.

**Project administration:** Cynthia Y. He.

**Resources:** Cynthia Y. He.

**Supervision:** Cynthia Y. He.

**Validation:** Htay Mon Aye.

**Visualization:** Htay Mon Aye, Feng-Jun Li, Cynthia Y. He.

**Writing – original draft:** Htay Mon Aye, Cynthia Y. He.

**Writing – review & editing:** Htay Mon Aye, Feng-Jun Li, Cynthia Y. He.

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
