## [Decision Letter · Decision Letter 0]

10 Jun 2024

Dear Dr. He,

Thank you very much for submitting your manuscript "Dynamic composition of stress granules in Trypanosoma brucei" for consideration at PLOS Pathogens. As with all papers reviewed by the journal, your manuscript was reviewed by members of the editorial board and by two expert independent reviewers. In light of the reviews (below this email), we would like to invite the resubmission of a significantly-revised version that takes into account the reviewers' comments.

Although both reviewers felt the paper had some interest, they also had substantial criticisms. One in particular Both were worried about cell viability, and one stated that at least some of the results might be due to a technical artefact - the fact that fluorescent proteins can be detected from within granules, whereas antibodies detect onl proteins on the outside (and therefore would also be less sensitive). Before this manuscript can be reconsidered it is thereofre essential that you address all of the questions raised by the reviewers. This includes providing the requested additional data on cell viability, and experiments in which the tags and antibodies are swapped between ALPH and PABP2. The possible technical issues must also be described and all additional methodological details requested provided. The Introduction needs to be re-written with clearer distinctions made between observations made in Opisthokonts and what is seen in trypanosomes.

We cannot make any decision about publication until we have seen the revised manuscript and your response to the reviewers' comments. Your revised manuscript will be sent to reviewers for further evaluation.

Sincerely,

Christine Clayton

Academic Editor

PLOS Pathogens

James Collins III

Section Editor

PLOS Pathogens

Michael Malim

Editor-in-Chief

PLOS Pathogens

orcid.org/0000-0002-7699-2064

Your manuscript has been reviewed by two experts in the field and although both it had some interest, but also had substantial criticisms. Before this manuscript can be reconsidered it is essential that you address all of the questions raised by the reviewers. This includes providing the requested additional data on cell viability, and experiments in which the tags and antibodies are swapped between ALPH and PABP2 in order to be certain that the differences observed are not just a technical artefact caused by the fact that antibodies do not penetrate the granules. These technical issues must also be described and all additional details requested provided. The Introduction needs to be re-written.

Reviewer's Responses to Questions

**Part I - Summary**

Reviewer #1: In this work, the authors have looked into stress granules that they observed to form upon incubation of trypanosome cells in PBS or PBS with glucose, but not with PBS that contains proline and also not with PBS that contains physiological amounts of glucose.

The paper contains a range of novel observations: (i) PBS with low amounts of glucose (PBSg) causes SG formation, but PBS with proline does not (ii) PBS granules and gPBS granules have different features (ii) there are two “behaviors” of granule components, namely “PABP-like” and “DHH1-like”, referring on whether components would stay in the granules upon ATP depletion or not. (iii) granules have a substructure, with interior and exterior localized proteins.

These data give interesting insights into the types and dynamics of trypanosome stress granules. However, I do have concerns with this work. Many details that would be necessary for me to judge the conclusions are unfortunately missing. Some experiments miss important controls. I list the points below.

Reviewer #2: This manuscript from Aye and colleagues describes changes in protein composition of stress granules (SG) in the African trypanosome, Trypanosoma brucei in response to changes in cellular ATP levels. Through a variety of microscopic approaches after manipulation of environmentally available carbon sources, the authors describe dynamic changes in SG protein membership. Further, expansion microscopy and STED have revealed that SG components localize to different portions of the granules. These are new findings in trypanosomes; however, the connection between ATP and SG formation has been well-established for some time in other systems, reducing the potential overall impact of these findings.

**Part II – Major Issues: Key Experiments Required for Acceptance**

Reviewer #1: Major points

• The introduction is too short and does not include any details on trypanosomes, about mRNA metabolism in trypanosomes, in particular, what is known and unusual about mRNA decay, translation, RNA granules. Translation initiation factors (why did the authors choose eIF4E1 rather than eIF4E3 in this work?)

• The same applies to Material and Methods: it lacks details on how the immunofluorescence was performed, which antibodies were used for the expansion microscopy and all other microscopy applications? How was the tri-marker cell line imaged, when it contains two fluorescent tags and one epitope tag? Expansion microscopy: what is the source of this protocol, that seems very different to standard protocols? How was the number of SG-containing cells quantified? (how do you define “SG-containing”?). Details on RNAi, image analysis, STED and Pearson’s coefficient are missing as well. It may seem not important on first view, which antibodies have been used and how exactly the imaging was done, BUT: the authors look into phase-separated areas which are restricted to antibody access, in particular this is true for the interior of trypanosome starvation stress granules, https://elifesciences.org/reviewed-preprints/95028: IgG antibodies, in particular anti-HA, stain only outside of the granule, while fluorophores are detectable throughout the granules. Therefore, it is important to know exactly how the detection of each protein was performed.

• Line 188: Why would ALPH1 be a canonical PB marker, while DHH1 is not? Both ALPH1 and DHH1 localise to both PBs and SGs ... If at all, DHH1 is slightly “more canonical”, because it does not localize to the posterior pole and is an established P-body marker in yeast.

• Cell death vs specific effect: Some experiments lack important controls, to test whether cells are still alive under the treatments, e.g. are the treatments reversible, when cells are retransferred into normal culture medium? One example is in Line 202 (another one is Figure 2D-F): ATP depletion by ED or 2DG prevents subsequent PB formation. How do the authors know, that these cells were still alive and not merely killed by the ATP depletion?

• Figure 5: DHH1 RNAi. There are no information on how the RNAi was done (Material and methods or Figure legend). No controls for the success of the RNAi are shown or mentioned. No information on when these images were made (how long after induction of the RNAi?). The image quality is not good. It is not explained, how many cells were analysed and how exactly the Pearson’s coefficient was calculated (parameters).

• Expansion microscopy data: These data are very nice and its an interesting observation. However, the differences could also be caused by different abilities of the antibodies to penetrate to phase separated areas, see comment above. Therefore, it is essential to add these details. It is also essential to swap the tags and antibodies between ALPH and PABP2.

Reviewer #2: 1. In Fig. 1A., the loss of ATP could result from cell death. While I appreciate that the literature supports that trypanosomes are viable in PBS for the study period, some evidence that they remain viable throughout the manipulation described here would be comforting.

2. In Fig. 1B, PABP2, ALPH1, and DHH1 in “Fed” cells “showed cytoplasmic distribution”. One possible explanation for this is that overall expression levels fall upon starvation, revealing foci of label. This should be assessed. Related, the text mentions what happens to the three markers at 30 min, but images are not provided. Please include those in Fig. 1B.

3. Fig. 1 C/D, but others also (Fig. 2D/F, Fig. 5B, Fig. 7B) – The legends need to include information about how many cells/ fields of cells were used to calculate the data used in these graphs to support the rigor of the work.

4. The experiments describing sub-granule localization need some clarification. First, the demonstration that PABP2 is in SGs by expansion microscopy is supported by co-localization with mRNA (in a supplemental figure). This localization is different (qualitatively – less punctate, fewer individual granules) than co-labeling with other markers. While I appreciate that the SGs are organized based on LLPS and therefore are inherently subject to possible morphological changes due to sample handling, is it possible that the granules described throughout this work are a mix of SGs and some other type of granule? The STED images of dT co-localization also has structures that do not label with dT. Some discussion of this issue is warranted.

5. The expansion images should include a brightfield image to allow assessment of cell quality. Please also include some indication of the percentage of granules that have the PABP2 labeling phenotype (hollow region at center or off to one side).

**Part III – Minor Issues: Editorial and Data Presentation Modifications**

Reviewer #1: • Figure 3 and S4: I would prefer to see the S4 data in the main Figures, as this is a key finding.

• Figure S5 is a major finding on the difference between PBS and gPBS granules, best not to hide in supplementary figures. The data show major differences in granule stability, that are very interesting.

• For DEAD box RNA helicases, the ATP is in general needed to release the enzyme from the RNA, rather than for the helicase activity itself, please  include this in the discussion -  why, then, should lack of ATP cause Dhh1 to be released from the SGs? (You would expect the opposite.)

• Line 306: “Indeed, DHH1 interaction with both ALPH1 and PABP2 is observed in Cary 2015 and Kramer 2023)”. Cary 2015 is a paper about DHH1 in yeast, which has no homologue to ALPH1. Same on line 373: ALPH1 is specific to Kinetoplastida and not present in yeast.

Reviewer #2: 1. Minor, but the graphs in Fig. 2A and B should be switched – PBS starvation is mentioned first in Fig. 1, so to be consistent it should likewise be first here.

2. Line 16, “remove the semicolon

3. Line 20, change “causing” to “that causes”

4. Line 35, add “the” before “bulk”

5. Line 39, remove the comma

6. Add “the” before “novel”

PLOS authors have the option to publish the peer review history of their article (what does this mean?). If published, this will include your full peer review and any attached files.

Reviewer #1: No

Reviewer #2: No
---

## [Decision Letter · Decision Letter 1]

7 Oct 2024

Dear Dr. He,

Thank you very much for submitting your manuscript "Dynamic composition of stress granules in Trypanosoma brucei" for consideration at PLOS Pathogens. As with all papers reviewed by the journal, your manuscript was reviewed by members of the editorial board and by several independent reviewers. The reviewers appreciated the attention to an important topic. Based on the reviews, we will accept this manuscript for publication, providing that you modify the manuscript according to the review recommendations. One reviewer has identified a small number minor issues that can be addressed by changing the text. Otherwise the reviewers were very satisfied by the way in which you have addressed their concerns.

Sincerely,

Christine Clayton

Academic Editor

PLOS Pathogens

James Collins III

Section Editor

PLOS Pathogens

Michael Malim

Editor-in-Chief

PLOS Pathogens

orcid.org/0000-0002-7699-2064

One reviewer has identified two minor issues that can be addressed by changing the text. Otherwise the reviewers were very satisfied by the way in which you have addressed their concerns.

Reviewer Comments (if any, and for reference):

Reviewer's Responses to Questions

**Part I - Summary**

Reviewer #1: The authors have addressed all the issues I had, in particular, the switch of the antibodies was very important and it's nice to see that it has worked and supports the findings. It is a very nice paper and I recommend publication.

Reviewer #2: This manuscript from Aye and colleagues describes changes in protein composition of stress granules (SG) in the African trypanosome, Trypanosoma brucei in response to changes in cellular ATP levels. Through a variety of microscopic approaches after manipulation of environmentally available carbon sources, the authors describe dynamic changes in SG protein membership. Further, expansion microscopy and STED have revealed that SG components localize to different portions of the granules.

**Part II – Major Issues: Key Experiments Required for Acceptance**

Reviewer #1: (No Response)

Reviewer #2: The authors have largely addressed my concerns. One that was not addressed – (from initial review, related to Fig. 1B) “In Fig. 1B, PABP2, ALPH1, and DHH….the text mentions what happens to the three markers at 30 min, but images are not provided. Please include those in Fig. 1B.” If the data is going to be mentioned, it should be included. Alternatively, the text could be altered to compare 0.25 h in the starved to 0.5 h in the gPBS, I think the conclusions would not change dramatically.

**Part III – Minor Issues: Editorial and Data Presentation Modifications**

Reviewer #1: (No Response)

Reviewer #2: 1. Author summary. Line 37, “In the single cellular Trypanosome parasites” to “In the single-celled trypanosome, SGs”

2. Line 74 – add “the” between “at” and “post-transcriptional”

3. Line 78. End the sentence after eIF4G. Then , “Additionally,….”

4. Line 369 – change “is” to “was”

PLOS authors have the option to publish the peer review history of their article (what does this mean?). If published, this will include your full peer review and any attached files.

Reviewer #1: **Yes: **Susanne Kramer

Reviewer #2: No

Figure Files:

Data Requirements:

Reproducibility:

References:

---

## [Editor Report · Decision Letter 2]

14 Oct 2024

Dear Dr. He,

We are pleased to inform you that your manuscript 'Dynamic composition of stress granules in Trypanosoma brucei' has been provisionally accepted for publication in PLOS Pathogens.

Best regards,

Christine Clayton

Academic Editor

PLOS Pathogens

James Collins III

Section Editor

PLOS Pathogens

Michael Malim

Editor-in-Chief

PLOS Pathogens

orcid.org/0000-0002-7699-2064
---

## [Editor Report · Acceptance letter]

25 Oct 2024

Dear Dr. He,

We are delighted to inform you that your manuscript, "Dynamic composition of stress granules in Trypanosoma brucei," has been formally accepted for publication in PLOS Pathogens.

Best regards,

Michael Malim

Editor-in-Chief

PLOS Pathogens

orcid.org/0000-0002-7699-2064